# The Role of Organizational Culture and Climate for Well-Being among Police Custody Personnel: A Multilevel Examination

**DOI:** 10.3390/ijerph18126369

**Published:** 2021-06-11

**Authors:** Christopher Robert Mark Werner-de-Sondberg, Maria Karanika-Murray, Thomas Baguley, Nicholas Blagden

**Affiliations:** 1School of Psychology, Social and Behavioural Sciences, Faculty of Health and Life Sciences, Coventry University, Priory Street, Coventry CV1 5FB, UK; ac7495@coventry.ac.uk; 2Department of Psychology, Nottingham Trent University, 50 Shakespeare Street, Nottingham NG1 4FQ, UK; thomas.baguley@ntu.ac.uk (T.B.); Nicholas.Blagden2@ntu.ac.uk (N.B.)

**Keywords:** UK police custody, well-being, organization culture and climate, multilevel analysis

## Abstract

United Kingdom Police custody is one of the most challenging of work environments, liable to excessive demands and reduced well-being. Being difficult to access, it is also a much-neglected area of research that has focused on one or two roles, rather than the full range available, and on individual-level research, rather than a more comprehensive multilevel understanding of how organizational culture and climate can simultaneously influence a range of well-being outcomes. The present longitudinal study explored all types of roles, in both the public and private sectors, across seven English police forces and 26 custody sites (*N* = 333, response rate 46.57%, with repeated returns = 370). The Integrated Multilevel Model of Organizational Culture and Climate (IMMOCC) was applied to examine the organizational-level influences on individual well-being. Results indicated that (1) custody sergeants were most vulnerable to low well-being, followed by publicly contracted detention officers; (2) shared leadership (a source of team cohesion) was linked to four of six well-being outcomes; (3) two sub-components of culture reflected tensions never acknowledged before, especially in respect of role; and (4) reverse relationships existed between well-being outcomes and the dimensions of culture and climate. The findings inform practical recommendations, including resilience training and the need to raise the status of police custody, while also highlighting concerns about private sector scrutiny that may be relevant to other professions.

## 1. Introduction

While general policing in England and Wales has been viewed as more stressful than specialist departments like police custody, the opposite is true when it comes to police sergeant custody officer burnout [1,2,3]. However, the tendency to focus on police sergeant custody officers overlooks the fact that that they are one of three (sometimes four) police custody roles that include both officer and police staff, where the latter can be publicly or privately contracted (i.e., Custody Inspector/Manager, publicly contracted; Detention Officer, publicly and privately contracted; and, occasionally, Custody Officer Assistant, publicly contracted). This mix produces two types of working relationship: (1) Officers and police staff are all publicly contracted and (2) officers are publicly contracted, while police staff are privately contracted. The type of role, contract, and subsequent organizational climate are important, albeit much-neglected, factors in research into police custody personnel work experience and, specifically, their well-being. In addition, working relationships vary depending on whether the custody units are stand-alone or located within police stations. These sites are also managed differently with stand-alone being self-sufficient in terms of management and staff, whereas police stations are not. This study sought to address the much-neglected exploration of the two combinations of role and contract and their implications for staff well-being in ways that will also take account of organizational culture and climate.

### 1.1. Expanding Police Custody (Officer and Police Staff) Perspectives

An initial attempt to address this was a relatively recent multilevel exploratory study on the well-being of custody sergeant and detention officers. The study’s results contradicted previous comparisons between health and criminal justice professions, which had suggested that practices conducive to staff well-being would be more common in the private, rather than the public, sector. Instead, the results showed considerable disquiet among private detention officers, which was linked to higher levels of emotional exhaustion and lower levels of personal accomplishment than their publicly contracted colleagues [4]. The results are intriguing, but the limitations of the study’s cross-sectional design and relatively small sample mean that they should be explored further. 

Therefore, further research is needed that is: (1) longitudinal, to better understand issues of police custody (officer and police staff) well-being over time; (2) multilevel, in order to account for the effects of cross-level links between individuals and groups; and (3) uses different outcome variables, in order to more fully understand the complexities of police custody staff well-being. The last is especially important for avoidance of conflating well-being with burnout (which is risked by negatively worded emotional exhaustion, as outlined in [5]). Some of these limitations are linked to several unresolved gaps related to contrasts between police custody (officer and police staff) role/sector well-being. Therefore, the first hypothesis is:
**Hypothesis 1** **(H1).***Privately contracted detention officers will report better well-being than publicly contracted police custody (officer and police staff) roles.*


There is an important difference between individual and group well-being, with the latter being conceptualized as an imbalance between power and control, which then undermines officer and police staff empowerment and trust [6]. This ‘balance’ is akin to leadership as a process in which everyone actively participates to produce a strong sense of team cohesion [7], also known as shared leadership. Shared leadership is defined as “people united in a common enterprise who share a history and thus certain values, beliefs, ways of talking, and ways of doing things’’ [6] (p. 243). It has been likened to organizational culture because, like organizational culture, it describes the ‘why’ of organizational behavior reflected in the deep-seated history of the organization, its policies, practices, and procedures [8]. It is in contrast with organizational climate as the ‘what’ of organizational culture, in terms of the meaning that employees attribute to events, policies, practices and procedures, and the behaviors they see expected, supported, and rewarded. As distributed leadership theory indicates, a balance between authority and power (i.e., control) is a key for promoting trust, motivation, accountability, and participation [9] (p. 367). Thus, organizational climate can be described as control belief climate (see Figure 1). Lacking empirical validation of shared leadership [9] (p. 357), the second hypothesis is:
**Hypothesis 2** **(H2).***Shared leadership positively predicts well-being outcomes.*

### 1.2. Multilevel Modelling of Police Custody (Officer and Police Staff) Well-Being

The need for a multilevel understanding was first addressed in [4], which first developed the Integrated Multilevel Model of Organizational Culture and Climate (IMMOCC; Figure 1). IMMOCC’s central structure is influenced by the Theory of Planned Behavior (TPB; [10,11]), with the difference that TPB is applied mostly in single rather than multilevel research. TPB is viewed as offering strong foundations for a multilevel model that can explain organizational culture and climate for well-being, especially in police custody officers and staff. Specifically, TPB’s indirect/direct outcome beliefs, normative beliefs, and control beliefs are conceptually close to distinctions between organizational culture (i.e., outcome and normative expectations; [12,13,14]) and control belief (organizational) climate [13,14], thus allowing us to predict outcome and outcome intentions. Its small number of predictors also make TPB immensely parsimonious [15]. By focusing on different levels of analysis, the multilevel advantage that IMMOCC offers is that it is able to measure culture at the shared level (level 2) and climate at both the shared and individual levels (levels 2 and 1), and also as indirect and direct influences on outcomes. In this investigation, outcome refers to well-being, operationalized as a multidimensional concept, as described below.

IMMOCC is briefly outlined for the purpose of developing the hypotheses, while the reader is also referred to [16] for a more detailed description. From left to right, Figure 1 shows shared/individual leadership as a background factor that impacts on organizational culture, which is operationalized as well-being belief culture and normative belief culture. These have both indirect and direct aspects. Individual leadership is aligned with organizational climate, which is operationalized as well-being control beliefs. Again, these have both indirect and direct aspects. In the TPB, their counterparts are called attitudes to well-being, subjective well-being norms, and perceived well-being control. In turn, these factors inform behavioral and/or psychosocial intention, and well-being outcomes. Perceived well-being control also predicts well-being outcomes directly, but only to the extent that it provides a proxy for actual well-being control (represented by a dashed line because it is also expected to be stronger when intention is a weak predictor of outcomes). Finally, the model incorporates reverse relationships whereby culture and climate are also outcomes of well-being (as well as being able to affect well-being) and climate can potentially lead to culture change [14]).

It is important to define and operationalize the well-being construct. This study takes the multidimensional view of the World Health Organization (WHO) definition of well-being, i.e., as a state in which every individual realizes their potential, copes with the normal stresses of life, works productively and fruitfully, and contributes to their local community [17]. This definition concurs with well-being as “a dynamic state in which the individual is able to develop their potential, work productively and creatively, build strong and positive relationships with others and contribute to their community” [18], adding that “it is enhanced when an individual is able to fulfil their personal and social goals and achieve a sense of purpose in society” (p. 10).

As such, well-being is a behavioral and/or psychosocial goal. It is behavioral in terms of working productively and fruitfully/creatively and able to contribute to one’s community. It is psychosocial, in terms of realizing/developing one’s potential, coping with the normal stresses of life, and building strong, positive relationships, and both behavioral and psychosocial, in terms of fulfilling personal and social goals and achieving a sense of purpose in society. It is a goal in terms of attainment being dependent on multiple outcomes, behavioral and/or psychosocial, the ‘balance’ of which is likely to contribute to well-being [15,19,20]. An example would be a custody sergeant who can cope with the excessive pressures of work, provided balance is at hand in the form of a good home life (or vice versa), only becoming unwell when that balance is denied because both are too difficult to cope with.

Additionally, note that organizational culture is multidimensional, with two sub-components: outcome expectations, normative expectations, and a third component of control belief climate (see Figure 1). Schein [12] defined organizational culture, as “a pattern of shared basic assumptions learned by [an organization] as it solved its problems of external adaptation (outcome expectation) and internal integration (normative expectation), which has worked well enough to be considered valid and, therefore, to be taught to new [staff] members as the correct way to perceive, think, and feel in relation to those problems” (p. 18; with parentheses added). These sub-component distinctions were also described by [13] and elsewhere as ‘values in action’ and ‘shared norms’ [8].

It is also important to note that culture has generally been approached as a singularly unidirectional construct, i.e., being either positive or negative [8], with no recognition that the two coexist and might work in opposition to each other [21]. In research on affective climates (e.g., service, safety, justice, inter-personal relations, efficiency, control, support, etc.) the two are expected to coexist [8,22,23]. From an expectancy-value perspective (as informed by TPB), outcome belief and normative belief can be expressed as strength of well-being/vicarious expectations and/or outcome evaluations/motivation to comply [15,24,25]. Examples of well-being (outcome) expectations [26] are the custody sergeant competencies of decision-making, leadership (i.e., leading change/people and managing performance), professionalism, public service, and working with others [27]. Examples of strength of vicarious (normative) expectations [26] are the people/task-focused aspects of organizational culture [28]. IMMOCC’s structure concludes with reverse inter-relationships, whereby well-being predictors are just as likely to present as outcomes. Given that reverse relationships were not considered in [4], this closes a gap concerning IMMOCC’s reverse relationships, where an interrelationship is expected between the cultural sub-components, such that:
**Hypothesis 3** **(H3).***Well-being belief culture and normative belief culture (indirect and direct) will influence each other.*
**Hypothesis 4** **(H4).***Reciprocal relationships exist between well-being outcomes and culture predictors of well-being, such that well-being outcomes (role well-being, well-being stress, mental and subjective well-being, energy, and engagement) influence predictor variables (well-being belief culture, normative belief culture, and control belief culture).*


## 2. Materials and Methods

### 2.1. Design

This study took a longitudinal, multi-strategy approach and incorporated quantitative, qualitative, and mixed (i.e., single, and multiple case study) elements. The research was conducted between Autumn 2015 and Spring 2017. The study presented here is largely quantitative with a brief qualitative focus in the Discussion section.

### 2.2. Participants

Seven English police forces agreed to take part, comprising: (1) two forces whose officers and police staff were all publicly contracted (nine units); (2) two forces whose officers were all publicly contracted and police staff privately contracted (seven units); and (3) three forces whose officers were all publicly contracted and police staff privately contracted, but whose contractor declined to allow their detention officers to take part (nine units)—a level of access granted, in part, due to the first author’s previous role as a police sergeant custody officer. Initially spread across 32 custody units (including one dummy coded for custody inspectors/managers, this reduced to 26 following new build replacement of old sites from the second wave onwards). Table 1 provides the participant demographics.

In terms of ethnicity, custody officers and police staff were largely Anglo/Celtic although the work environments were culturally quite diverse. This reflected the geographic areas of the forces, with one being a metropolitan force, two forces being largely rural and of low population density, two forces that sat between these two extremes, a sixth that had a higher-than-average mixed-race community than anywhere in the country (i.e., black and minority ethnic, plus migrant, refugee, and asylum-seeker populations), and the last reflected a population that was on average older compared to the rest of England.

### 2.3. Materials

Item development was based on three considerations. First, levels 1 and 2 necessitated qualitatively different composition models [30], i.e., direct consensus for individual levels and referent-shift consensus for shared levels. Second, all predictors (except for well-being intentions) were measured both indirectly and directly to account for different assumptions about individual abilities to access and report them [24]. Finally, the ‘principle of compatibility’ was taken into account, whereby predictors of an outcome are said to be compatible if their target, action, context, and time are assessed at the same level of generality or specificity [11,15,24]. For the current research, this concerned well-being as outcome (target), predicted by numerous (action) statements, regarding the participant working in police custody (context) over the next six months or so (time). This saw referent-shift compositions take on ‘if/then’ introductions similar to: “Taking into account the extent to which you see statements as true or false, how likely is it that they will contribute to your team/work group achieving work-related well-being over the next 6 months or so.”

All scales were averaged following item deletion to maximize omega reliability [31]. Intraclass correlation coefficients were used to allow for weak ICC1s and strong ICC2s [32]. Aggregation is an emergent process whereby the individual and shared levels are related but not isomorphic [32,33]. ICC1s of 0.15–0.68 are typical of small group research [4,34] while ICC1s ≥0.1 and group sizes >15 justify multilevel analysis [35].

Item reduction of new measures that were bespoke and adapted for multilevel use was achieved using exploratory factor analysis (EFA). New items were created for shared leadership, well-being intention, and role well-being (all based on extant literature; see below). Measures were improved on [4] for parsimony [24,25,26]. It was also important to reduce ambiguity of measures used in an Australian prison population [36] to apply them to a police custody population in England.

EFA was chosen over confirmatory factor analysis (CFA) for two reasons: first, due to the absence of a priori expectations about the number and influence of common factors necessary for CFA; second, because existing theory and information supporting adapted measures offered little or no insight into the current research [37]. Analyses were conducted using principal axis factoring with Varimax rotation, Kaiser–Meyer–Olkin (KMO) measure of sampling adequacy, communalities, determinant, Bartlett’s test of sphericity, anti-image correlation [38], and critical value loading options [39] (*N* = 330, with no repeated data). The original 163 items were gradually reduced to 114 items at the final wave (*N* = 367). Full details of the survey development are available from the first author.

Table 2 contains the summary of predictor, outcome, and control variables, with sample items contained in Appendix A, Table A1.

### 2.4. Procedure

Full ethics approval was granted by the Research Ethics Committee of Nottingham Trent University. This was a four-wave panel study with a lag between surveys of five months. Participants were informed of the voluntary nature of the research, right to withdraw, and informed consent. To link repeated returns while preserving anonymity and confidentially, participants were asked to provide a unique identifying code using a specific formula if they so wished (based on their birthday and month and last three digits of their primary telephone number). Participants had the option to complete the survey either on paper (returned to the researchers in a pre-paid envelope) or as an online survey. All staff received advanced notice of the research from their Heads of Department and a written invitation to the research that also affirmed that anonymized aggregated data would be fed back at the end of each survey and across all four surveys.

### 2.5. Data Analyses

H1 was tested using a single, one-way, between-groups ANOVA with 1000 (bias corrected accelerated; BCa) bootstrapped samples and Hochberg post hoc corrections at *p* < 0.01 [39], which allowed to control for overall and per test Type 1 error rate. The data showed a violation of independence due to the multilevel nature of the sample (i.e., individuals nested within teams) and the correlated nature of the scales/use of repeated measures. Homogeneity of variance was seemingly violated due to the group sizes being unequal (largest/smallest ratio of variance was 8.5, larger than the recommended 1.5), although the non-significant Levene tests and similarity of standard deviations in [16] refuted this.

H2 was tested using hierarchical linear modelling’s random coefficient approach [48], where outputs from three different models build on each other [49]. This starts with the null model, absent of individual and shared-level fixed effects, progresses to the individual-level fixed effects and covariate demographics model, and ends with the shared-level fixed effects model controlling for model two and the potential for team differences. Since the focus here was on shared leadership, only results from the third shared-level fixed effects model will be presented, using two-tailed tests, and reporting 95% and 90% confidence intervals, as appropriate, for directional predictions.

H3 was tested using the same approach as for H2 but only concerned with direction rather than statistical significance.

Finally, H4 was tested using the same approach as for H2. Here, again, third model focus was on parsimony while using two-sided tests for directional predictions.

In addition, participants’ open comments at the end of the survey were analyzed using thematic analysis [50]. These informed the interpretation of the findings (in the Discussion section) but are not presented here (please contact the first author for more information).

## 3. Results

### 3.1. Sector and Role Comparisons (H1)

Results produced statistically significant (or marginally significant) differences for sector and role (combined as a single variable) for all outcomes except role well-being. In descending order, these were low workplace stress F(4, 336) = 10.18, *p* = 0.0005, ƞₚ² = 0.01 (a small effect size); mental well-being F(4, 336) = 4.66, *p* = 0.001, ƞₚ² = 0.05 (approaching a medium size effect); energy F(4, 336) = 3.94, *p* = 0.004, ƞₚ² = 0.05 (approaching a medium size effect); subjective well-being F(4, 336) = 3.61, p = 0.007, ƞₚ² = 0.04 (a small to medium size effect); and engagement F(4, 336) = 2.80, *p* = 0.026, ƞₚ² = 0.03 (a small effect size). In addition, well-being intention F(4, 336) = 2.00, *p* = 0.095, ƞₚ² = 0.02 (a small effect size) was included for completeness. Mean comparisons/post hoc differences between private detention officers and other sector roles are summarized in Figure 2.

Differences were statistically significant for custody sergeants in five of the six well-being outcomes and for publicly contracted detention officers in two of the outcomes. Therefore, results provide partial support for H1. It is worth pointing out that while custody assistants (*n* = 17) enjoyed the best well-being overall, privately contracted detention officers (*n* = 63) provided the more reliable result. Findings for privately contracted detention officers were very different to those in [4], possibly due to the appointment of a new contractor.

Note that there was evidence for high levels of pervasive negative affectivity (NA), which was initially employed as a control measure but also produced substantive effects [51,52,53]. The data demonstrated that relatively low mean raw scores for NA can quickly translate to high levels (expressed as percentiles), with substantial impact for all police custody roles (except for custody assistants). In addition to specific role vulnerabilities, over 25% of all police custody (officer and police staff) roles had a raw score of ≥20 and NA percentile of 81, over 10% of police custody (officer and police staff) roles had a raw score ≥25 and NA percentile of 91, and nearly 2% of police custody (officer and police staff, but not custody assistants) roles had a raw score of 35 to 50 and NA percentile >99.

### 3.2. Shared Leadership and Team Cohesion (H2)

Shared leadership, an example of police custody (officer and police staff) team cohesion, positively predicted well-being outcomes. H2 was supported for subjective well-being, t(24) = 2.53, 95% CI [0.08, 0.66], r = 0.46 (approaching a large size effect); engagement, t(24) = 2.05, 95% CI [0.01, 0.29], r = 0.39 (a medium size effect); and mental well-being t(24) = 1.76, 90% CI [0.01, 0.16], r = 0.34 (a medium size effect). Although energy, t(26) = 1.60, was statistically non-significant, it supported shared leadership at the individual level, t(26) = −1.97, 90% CI [−0.34, −0.03], r = 0.42 (a medium to large size effect). This provided strong support for H2 in respect to four of the six well-being outcomes, suggesting good levels of team cohesion.

### 3.3. Culture Sub-Component Influences (H3)

H3 focused on the co-existence of cultural sub-components, which could, potentially, work in opposition to each other. H3 was not supported for subjective well-being or engagement but was supported for the remaining outcomes. Specifically, it was supported for role well-being, where normative belief culture (indirect) was a positive predictor, while attitude to well-being (direct) was a negative predictor. It was supported for energy, where normative belief culture (indirect) was a positive predictor, while attitude to well-being (direct) was a negative predictor. It was also supported for low workplace stress, where normative belief culture (indirect) was a negative predictor, while attitude to well-being (direct) was a positive predictor. Finally, it was supported for mental well-being, where both well-being and normative belief culture (indirect) were positive predictors, while attitude to well-being and subjective norm (direct) were negative predictors.

While the evidence of coexistence among the cultural sub-components (positive and negative) was clear for role well-being, energy, and workplace stress, the results for all four are difficult to explain. The results for role well-being, energy, and workplace stress suggested a potential for tension between the sub-components, which had never previously been observed or acknowledged but which has unknown quantitative and/or qualitative implications. Hence, support for H3 was partial.

### 3.4. Reverse Support for IMMOCC (H4)

Support for H4 was provided (see Table 3). Specifically, mental well-being predicted all four indirect measures of the predictors. Negative affectivity, attitude to well-being, and engagement/disengagement predicted three indirect measures of the predictors, while nine other outcome variables predicted two predictor variables, as depicted in the IMMOCC model (Figure 1).

To summarize, 21 out of 24 variables provided meaningful, if not statistically significant, findings, demonstrating support for reverse relationships between well-being outcomes and culture/climate predictors (H4). Of interest were the following. (1) Variable shifts predicted shared leadership; (2) engagement predicted shared leadership and normative belief culture; (3) low intolerance for ambiguity predicted shared control belief climate; (4) 12-h shifts and intolerance for ambiguity predicted normative belief culture; and (5) disengagement predicted shared control belief climate. These results suggested two contrasting custody environments. Specifically, variable shifts (9–11 h) were epitomized by a shared leadership culture and a more tolerant/engaged climate, while 12-h shifts were epitomized by a culture and climate that was less tolerant and more disengaged.

The strength of cultural sub-component tensions for role was unexpected. In addition, shared leadership and well-being belief culture were also generally far more positive than the completely negative normative belief culture (see Table 4). These results are a stark contrast to H3, since H3 focused on well-being outcomes, whereas H4 focused on shared leadership, sub-components of culture, and climate as former predictors.

## 4. Discussion

### 4.1. Summary of the Results

The first three hypotheses focused on the linear aspects of the model, which can be summarized as well-being, theory, and method. Specifically, it was found that custody sergeants had the lowest well-being followed by public detention officers, while custody assistants enjoyed the highest well-being followed by private detention officers (H1).

Note, however, that there was evidence for high levels of pervasive negative affectivity (NA), highlighting the vulnerability of all police custody roles (officer and police staff) more generally, but also specifically concerning custody sergeants (whose scores were below the mean), publicly contracted detention officers (with scores above the mean), and privately contracted detention officers (who occupy the mean and, thus, remain vulnerable to poor well-being despite new contractor improvements). Hence, the current gap between public and private sector custody well-being may not be as great or as concrete as expected. For this reason, an evaluated training program to improve officer and police staff resilience across all sector roles is recommended. (We return to recommendations shortly.)

The present study evidenced the value of shared leadership as a basis for team cohesion across four of the six well-being outcomes (H2). However, distributed leadership theory [40] indicates that to conflate the two may undermine the conceptual rigor of distributed leadership and its ability to systemize multiple leadership perspectives. Nevertheless, the results are important for the fact that when looking to recommendations for improving custody well-being, they indicate a continued need to focus on multilevel relationships where individuals are nested within teams.

Contrasting work environments in terms of shift patterns (i.e., variable and 12-h shifts) showed the former as positive or highly conducive to well-being and the latter as negative or less conducive to well-being. This contrasted linear results for 12-h shifts that had generally been more favorable 16]. These findings are important because, when looking to recommendations for improving custody well-being, they highlight the need to promote positive work environments, which are epitomized by a shared leadership culture and a tolerant/highly engaged climate, and the need to be cautious of potentially negative environments, which are epitomized by an intolerant and disengaged culture and climate. Results do not decry the use of 12-h shifts but suggest caution over their blanket use, for example, in custody units situated within police stations (often smaller), which may be less able to provide breaks, and stand-alone larger custody sites, which may be better able to provide breaks [16].

These results are important because, when looking to recommendations for improving custody well-being, they highlight the need to raise the status of custody [16] to remove cultural sub-component tensions exemplified as culture change, staffing, and clarity. In terms of culture change, short-, medium-, and long-term change can be largely achieved by targeting climate factors [14]. For example, in the short term by ensuring custody ITS is fit for purpose and in the medium to long term so that custody sector roles are able to see that all levels and departments within the police speak positively about custody and that there are no conflicts among any of the roles. In terms of staffing, recommendations include avoiding adherence to minimum staffing, provision of breaks for officers, a sickness policy that is fit-for-purpose and adhered to (including the need for secondary risk assessment when returning officers and police staff to custody, having shown themselves unable to cope), and staff shortages/vacancies being filled at the earliest opportunity (with succession plans prioritizing the needs of custody so that local management do not cherry pick by offering up sergeants with poor sickness records). Finally, in terms of clarity, regarding the use of stand-alone, larger custody sites, ensure vulnerable officers have a choice to work at either a stand-alone unit or one situated within a police station (though always subject to secondary risk assessment); have a focus on 12-h shifts (as above); and work with the private sector to verify well-being across more than one contractor.

### 4.2. Theoretical Implications

This was the first empirical study validation of shared leadership (H2), which provides the basis for the importance of team cohesion (at least in four of six well-being outcomes). While linear support for IMMOCC was crucial, more enlightening was the unexpected strength of cultural sub-component tensions in the reverse relationships for role (H4), where shared leadership and well-being belief culture were generally positive while normative belief culture was completely negative. Overall, the results provided support for IMMOCC and the need to continue to explore the ways in which culture and climate impact on a range of well-being outcomes, over time, and reciprocally.

### 4.3. Critical Reflections

#### 4.3.1. Study Limitations

The findings should be interpreted in light of three potential limitations. First, workplace stress was measured by a single item. This was problematic because of its poor re-test reliability (i.e., second wave = −0.02; third wave = 0.06; and fourth wave = 0.07) and because it was omitted by 6.27% of the participants. The latter could be due to emotional inhibition [54,55] or a reluctance to admit emotional experience. It is proposed that the single item used to measure stress could be replaced with a multiple-item perceived stress scale (e.g., PSS-10 [56]).

Second, participants seemed unable to comprehend some of the multilevel ‘if/then’ item introductions. Although these were developed to comply with the principle of compatibility, they may have also resulted in potential ‘underestimation’ [57], with some participants providing only neutral answers and others providing no answers at all. However, most participants expressed no difficulty completing the survey, which is supported by evidence [58] that the study’s large sample skewness and kurtosis were insufficient to cause a threat to outcomes.

Third, conducting police custody research, vis-à-vis the highly stressful nature of the work, can be difficult. Some officers and police staff do not seem to cope well when working in police custody. This sample showed some difficulty completing ‘if/then’ items, possibly due to cognitive overload at times when cognitive ability is compromised due to an imbalance between arousal (i.e., stress, anxiety, and depression) and performance [54,55,59,60]. The potential excessive demand can outstrip one’s ability to cope, suggesting the possibility that a shorter survey could address cognitive load and facilitate participation. In addition, it was difficult to gain permission from private contractors for their staff to be surveyed (also experienced in an earlier exploratory study [4]). This limited the research to private detention officers represented by one contractor only, which led to a lack of private sector comparison.

#### 4.3.2. Placing the Research within the Wider Literature

Although the need to raise the status of custody has been mentioned, there is benefit in going back to a time when police custody outsourcing was in its infancy [61], because many issues of concern then also remain prevalent today (also [4]). Nevertheless, it is of concern that outsourcing has not changed the fact that custody sergeants remain predominately mature males whose role is considered far from prestigious. In addition, outsourcing is problematic in terms of staff retention and morale, as it offers only basic staff remuneration. In addition, outsourcing reduces public accountability because the private sector is not subject to the same level of scrutiny as the public sector [62,63] [4].Support for concerns around outsourcing can also be found in the role of custody nurses who are employed 24/7, while being supported by on-call police surgeons. An evaluation of their pilot introduction [64] mirrored the issues found in the present study.

These findings showed that ‘good’ police custody is something more than the use of soft skills [65]. Although beyond the direct aim of this study, it is safe to infer that the tensions identified by police custody officers and police staff in the current study could be resolved by creating a national police custody service staffed by police custody inspectors/managers and sergeants seconded for the purpose (as in other branches of the service like regional police training), who would continue to be supported by civilian detention officers/assistants and nurses. Allowing police custody to receive the specialist status that it deserves would also fulfil the need for evidence-based excellence in police custody [66], as the current research was intended to play its part.

### 4.4. Future Research Implications

Based on the findings, several future research implications can be offered for custody-specific research. First, as an initial empirical validation of shared leadership, the current study calls for further research. Second, there is a need to implement/evaluate training programs to improve officer and police staff resilience. Third, and building on the second limitation, it would be important to develop a shorter survey to assess custody staff well-being in terms of coping. Fourth, it would be important to better understand the context of the private sector and, specifically, to confirm the reliability of results on well-being and to conduct comparative research across multiple contractors, where possible. Here, the need for caution stems from having data from only one company whose results contrasted both positive well-being and high negative affectivity. However, given the difficulty of obtaining private sector permission to conduct this kind of research, further investigation may only be possible by enlisting help from the National Police Chiefs Council and/or Ministry of Justice.

Broader research would be important to conduct in two priority areas. The first area recognizes that the earlier recommendations (Section 4.1) could potentially also benefit staff–prisoner relations, particularly where officers and staff do not cope well working in police custody and, therefore, presenting difficulties in officer and staff–prisoner relations. This fact should be considered when implementing measures to improve police custody officer and police staff well-being. This parallels the HM Prison Service, where it was identified that differences between the private and public sectors regarding dimensions of culture, confidence in the use of authority, knowledge/experience, and the delivery of safe/reliable regimes can adversely impact staff well-being [67] and potentially staff–prisoner relations [68]. For police custody, this relational focus could be achieved by assessing whether changes in police custody staff well-being and resilience also affect staff–prisoner relations. The second area focuses on the need to conduct more and more comprehensive research on well-being among staff of the National Health Service (NHS) and the Prison Service, especially focusing on the often-neglected role of culture/climate on well-being. In the NHS, this is supported by headlines such as “Lack of social care is piling pressure on surgeries and A & Es [Accident & Emergency departments]” [69] and in the prison service as “Austerity cuts blamed for prison ‘crisis’” [70] or “Austerity has a negative impact on Europe’s prison services and disheartens staff” [71].

### 4.5. Study Impact and Strength

While IMMOCC still has room for improvement, the model has substantial utility and potential for explaining well-being in a variety of organizational settings. The model can also benefit larger organizations in terms of supporting staff competencies and the application of a multilevel approach to understanding how factors in the organizational context impact on individual outcomes. The weight of empirical evidence provided by this study supports this.

The study’s strength is in two areas. First, its multi-strategy design, of which only a part of the quantitative analyses has been presented, and second, its inclusion of a large and rich set of data from all five police custody roles, where all but one (i.e., custody officer assistant) were spread across seven police forces (including two involving privately contracted detention officers, albeit from the same contractor). This supports the generalizability of and confidence in the findings.

## 5. Conclusions

The present study examined the relationships between a range of organizational factors and well-being across several police custody sector roles. This is both a difficult-to-access group and a much-neglected area of research, which has typically focused on one or two rather than the full range of roles. It achieved this by establishing a clear context for well-being among police custody officers and police staff (public and private) in the form of IMOCC, representing the important role of organizational culture and climate for officer and police staff well-being holistically as the basis for a multilevel approach and developing recommendations for improving the context of police custody and beyond.

## Figures and Tables

**Figure 1 ijerph-18-06369-f001:**
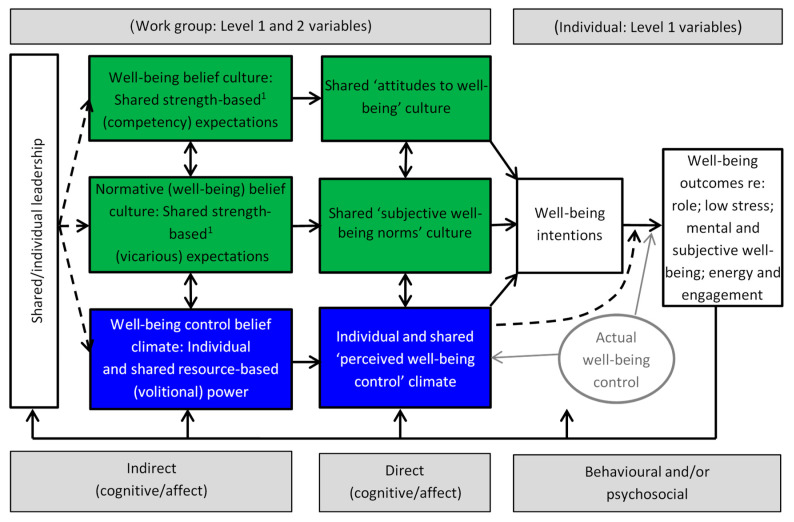
Integrated multilevel model of organizational culture and climate for police custody staff well-being. Note. ^1^ The model updates earlier version [4].

**Figure 2 ijerph-18-06369-f002:**
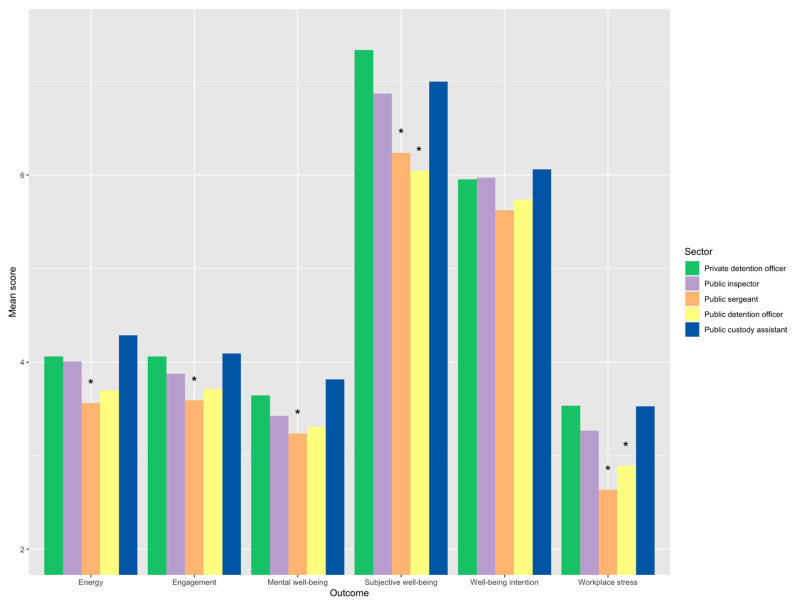
Clustered bar of mean scores for well-being outcome by sector and role with statistically significant differences indicated by *.

**Table 1 ijerph-18-06369-t001:** Participant demographics.

Role	Participants (*N*, %) & Repeated Returns (*N*)	Age (*M*, *SD*)	Tenure in Years (*M*, *SD*)		Males (%)	Full-Time (%)	Variable Shifts (%)
			Police	Custody			
Inspector	25 (83.33%), 30	47.00, 4.87	22.49, 4.49	2.6, 2.71	77.27	84.00	68.00
Sergeant	167 (46.52%), 189	44.49, 6.98	18.72, 5.78	3.94, 3.95	85.33	92.81	71.86
DO-Public	54 (30.17%), 60	44.72, 10.67	14.01, 5.43	12.88, 5.41	57.45	88.89	79.63
DO-Private	60 (87.5%), 67	34.43, 10.81	5.75, 5.21	5.30, 5.12	59.26	95.24	-
Assistant	17 (22.67%) ^1^	23.20, 3.73	0.96, 0.69	0.70, 0.60	64.29	82.36	100
Total	333 (46.57% ^2^) 370						

Note. *N* = Number. *M* = Mean. *SD* = Standard deviation. DO = Detention Officer. ^1^ Unknown = six with two only partly completed, though contributing to factor analysis. ^2^ A census approach aimed at return of 50% [29].

**Table 2 ijerph-18-06369-t002:** Summary of variables and their reliabilities.

Variable	*N*	Composition [30]	Omega Reliability (95% CI)	ICC2	ICC1 ^1^
**Predictors**					
Shared Leadership ^2^ [6,9,40]	5	Referent-shift	89 (0.87, 92)	0.89 (0.87, 91)	0.62 (0.57, 67)
Well-being belief culture ^3,4^ [4,26]	27	Referent-shift	0.85 (0.82, 88)	0.77 (0.72, 81)	0.45 (0.40, 51)
Normative (well-being) belief culture ^5^ [27]	11	Referent-shift	0.82 (0.77, 85)	0.74 (0.69, 78)	0.20 (0.17, 24)
Well-being control belief climate ^2,3^ [41]	9	Referent-shift	0.89 (0.87, 91)	0.88 (0.85, 90)	0.44 (0.39, 49)
Attitudes to well-being culture	2	Referent-shift	0.81 (0.86, 94)	0.91 (0.88, 92)	0.83 (0.79, 86)
Subjective well-being norms culture	2	Referent-shift	0.78 (0.74, 86)	0.80 (0.76, 841)	0.67 (0.61, 73)
Perceived well-being control climate ^2^	2	Referent-shift	0.93 (0.90, 95)	0.92 (0.90, 94)	0.85 (0.82, 88)
**Outcomes**					
Well-being intentions	7	Direct consensus	0.85 (0.81, 89)	0.84 (0.82, 87)	0.43 (0.39, 48)
Role well-being	7	Direct consensus	0.83 (0.78, 86)	0.82 (0.78, 85)	0.39 (0.34, 44)
Low workplace stress [42]	1	Direct consensus	N/A (see study limitations)		
Mental well-being [43]	7	Direct consensus	0.90 (0.88, 92)	0.86 (0.83, 88)	0.46 (0.42, 51)
Subjective well-being [44,45]	4	Direct consensus	0.87 (0.84, 89)	0.86 (0.83, 88)	0.60 (0.55, 65)
Energy [4,5]	8	Direct consensus	0.81 (0.76, 85)	0.71 (0.66, 75)	0.23 (0.19, 28)
Engagement [4,5]	8	Direct consensus	0.74 (0.67, 78)	0.72 (0.67, 76)	0.24 (0.20, 29)
**Controls**					
Negative Affectivity [46]	10	Direct consensus	0.87 (0.80, 91)	0.88 (0.85, 90)	0.41 (0.37, 46)
Intolerance for Ambiguity [36]	4	Direct consensus	0.85 (0.81, 88)	0.82 (0.78, 85)	0.53 (0.47, 58)

Note. *N* = Number. CI = Confidence interval. ICC = Intraclass coefficient. ^1^ Provides effect size [34,47] for influence of team membership. ^2^ Group mean-centered provides individual level for comparative purposes. ^3^ Sub-scales’ reliabilities were in the same range or higher. ^4^ Based on sub-scale means due to computational difficulties. ^5^ Uses the two people-focused factors of the OCI.

**Table 3 ijerph-18-06369-t003:** Fixed effect outcomes for indirect measures and predictor covariates.

	Shared Leadership			Well-Being Belief Culture			Normative Belief Culture			Shared Control Belief Climate		
	Coeff. ^1^	SE	t	Coeff. ^1^	SE	t	Coeff. ^1^	SE	t	Coeff. ^1^	SE	t
Role: Custody officer assistant ^2^	1.96 *	0.94	2.09							2.37 **	0.69	3.45
Role: Inspector	−1.22 *	0.67	−1.81									
Role: Sergeant							−0.63 *	0.27	−2.34	-0.61 *	0.35	−1.77
Role: Detention officer							−0.39 ^(tr)^	0.27	−1.44	−0.56 ^(tr)^	0.35	−1.60
Sector				−0.75 **	0.27	−2.75						
Contract	0.46 ^(tr)^	0.29	1.57									
Age				0.02 *	0.01	2.14	0.01 *	0.01	2.09			
Gender							0.19 *	0.11	1.75			
Tenure in police	0.03 ^(tr)^	0.02	1.44									
Shift hours	−0.42	0.25	−1.64				0.20 ^(tr)^	0.12	1.62			
Low negative affectivity				−0.15 *	0.07	−2.12	−0.15 *	0.06	−2.48	−0.15 *	0.08	−2.01
Low intolerance for ambiguity							−0.06 *	0.03	−1.99	0.05 ^(tr)^	0.04	1.40
Attitude to well−being	0.14 **	0.05	2.84	0.08 *	0.04	2.17				0.14 **	0.04	3.75
Subjective norms	0.09 ^(tr)^	0.06	1.62	0.08 ^(^*^)^	0.03	1.65						
Shared perceived climate							0.16 **	0.06	2.61	0.21 **	0.08	2.65
Individual perceived climate							−0.10 ^(^*^)^	0.06	−1.67			
Well-being intentions				0.17 *	0.08	2.09				0.14 ^(^*^)^	0.08	1.69
Role well-being							0.19 *	0.08	2.45			
Mental well-being	0.24	0.16	1.46	0.30 **	0.12	2.53	0.17 *	0.10	1.73	0.27 *	0.13	2.17
Subjective well-being										−0.07 *	0.04	−1.82
Engagement	0.16	0.11	1.40				0.14 *	0.07	2.12	−0.07 *	0.04	−1.82

Note. Coeff = Coefficient. SE = Standard error. t = *t*-test. ^(tr)^ Trend; ^(^*^)^ approaching statistical significance; * two-tailed *p* < 0.10; ** two-tailed *p* < 0.02. Notes: ^1^ Coefficients are unstandardized. ^2^ Included as dummy coded referent in intercept. All effects estimated using Full Maximum Likelihood estimation.

**Table 4 ijerph-18-06369-t004:** Cultural sub-component tensions.

	Shared Leadership ^1^			Well-Being Belief Culture			Normative Belief Culture		
	Coeff	SE	T	Coeff	SE	t	Coeff	SE	t
Role: Custody officer assistant ^2^	1.96	0.94	2.09	2.62	0.66	3.94	−0.37	0.53	−0.70
Role: Inspector	−1.22	0.67	−1.81	−0.97	0.44	−0.16	−0.51	0.33	−1.55
Role: Sergeant	−0.48	0.48	−1.00	−0.12	0.34	−0.36	−0.63	0.27	−2.34
Role: Detention officer	0.10	0.47	0.21	−0.46	0.34	−1.37	−0.39	0.27	−1.44
Summed totals	0.36	2.56	−0.51	1.07	1.78	2.05	−1.90	1.40	−6.03

Note. ^1^ Likened to organizational culture [6]. ^2^ Included as dummy coded referent in intercept.

## Data Availability

The data are available to the public at: https://osf.io/57hae/?view_only=1075b05bf21548ada3a129caacc2df48; https://doi.org/10.17605/OSF.IO/57HAE (accessed on 4 June 2021).

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
