# Peer review of "The Role of Organizational Culture and Climate for Well-Being among Police Custody Personnel: A Multilevel Examination"

_ijerph, 2021, doi:10.3390/ijerph18126369_

Round 1

Reviewer 1 Report

Although this is an under-researched area of study that has become a priority in security institutions, the impact of the study remains very limited. It would have been interesting for the authors to go beyond a descriptive presentation of the results obtained. For example, how to explain this gap between the public and the private? What does this mean in practice and how can we plan preventive policies in terms of well-being at work? In the current state, the manuscript brings little new knowledge, but apart beyond the methodological limitations of previous studies. I invite the authors to review the discussion and conclusion sections so as to make the results more "meaningful" from a point of view of comprehension and practical consequences.

In the summary, the hypotheses are presented more as research questions. The wording needs to be reviewed.

The method section needs to be reviewed. The presentation of the variables is very long and too detailed. In a thesis, it is appropriate to make such a presentation, but in a paper, a synthesis is expected. The use of a table could be useful.

Figures 3.2 and 3.3 are redundant, my preference would be to keep 3.3 which is more "meaningful". Authors would need to identify the differences with asterisks in the bar chart.

Table 4 should be included in the results section. It would also be necessary to better defend its relevance.

Finally, as explained, it would be appropriate to revise the discussion and conclusion sections so as to better highlight the contributions of the study.

Reviewer 2 Report

A well written and executed paper, although the subject is very specialized. Furthermore, the methodology seems a bit overworked compared to what the paper does. 

The present paper concerns the well-being of policy custody personnel in the UK. This is interesting as such but also has wider implications. While this is not explicitly stated in the paper, other industries also use a mix of public provision of services and subcontracted private services. The results in this paper thus also has a bearing on these industries.

For the actual topic of police custody personnel, the scientific contribution is obvious, as earlier literature only has focused on police sergeant custody officers. In this study, however, three more personnel roles within policy custody is studied, which is new. There has also been other development in police custody, such as modern custody sites are much larger than before.

The paper is very well-written, and well executed, with in-depth statistical analysis. If anything, the statistical analysis perhaps overdoes thing a bit, as the data set is not very large.

Reviewer 3 Report

Dear Authors,

This research address some significant issues regarding workplace experiences of custody and ultimately implications for peoples experiences in custody.

This research was clearly quite complex and extensive. I would have liked more context and information regarding the participants and the communities where they are employed. This would be helpful from an international perspective and would enable assessment for transferability across contexts.

Individual cultural background was not discussed in depth, and gives a sense of a largely Eurocentric approach to the research. Including this in your approach to workplace culture would add depth to this research. Or at least identify if participants were all from Anglo/Celtic backgrounds. 

This is clearly a contentious subject in current times with Black and Indigenous deaths in custody continuing. It would be important to acknowledge this within your background, clearly there is significance in this research related to theses issues. Are there links that can be made to negative experiences of custody related to your findings? 

Kind Regards 

Reviewer 4 Report

The authors aimed to examine psychosocial factors and the well-being of police custody personnel. However, overall, the manuscript is lengthy, off the point, not well-organized, and hard to read. It seems that the manuscript is based on Ph.D. research, but an academic paper is quite different from a Ph.D. thesis. The authors need to revise the manuscript considerably. For example, the insufficient explanation made me quite hard to judge how rigidly and precisely they analyze the data. In any case, the authors need to improve the overall clarity of the manuscript while shortening it.

Abstract: What are the main results and conclusions of this study? The authors need to describe them more plainly and clearly.

"Results address four hypotheses regarding:... " and "Conclusions set results in..": These are unusual expressions as a paper abstract. Please revise the abstract.

2.1. Design: Here, the authors should describe the study design, such as cross-sectional or longitudinal.

2.2. Participants: This section is messy. Please consider using a Table.

2.3. Materials – survey development: The authors reported that they used exploratory factor analysis. However, it has too many items considering the number of data. Also, the inclusion and exclusion criteria from a factor were not explained. Furthermore, the author did not mention which rotation method they used nor the acceptable KMO value. I doubt the adequateness of the survey. 

2.8. Data analysis: Here, explain all analysis used in this study, rather than simply mentioning "analyses were almost exclusively regression-based..." (L. 417).

Others: Basic information is lacking. For example, when was the study conducted? Also, the usage of 90% CIs and one-tailed p values are not recommended. Why did the authors not show effect sizes? The authors used too many parentheses.

Round 2

Reviewer 1 Report

The presentation of the results remains complex to read and I believe that the discussion should be revised to improve the writing. The appendices are voluminous, I invite the authors to reduce and limit themselves to relevant content. I would tend to present the assumptions in the summary. I also recommend a full linguistic revision of the text.

Author Response

The presentation of the results remains complex to read and I believe that the discussion should be revised to improve the writing. >>> Done, thank you.

The appendices are voluminous, >>> Two of the three Appendices have now been removed.

I invite the authors to reduce and limit themselves to relevant content. >>>Done, thank you.

I would tend to present the assumptions in the summary. >>>Summarised under sub-section 2.5. Data analyses.

I also recommend a full linguistic revision of the text. >>>> Done, thank you.

>>>Note. There are no longer any colour coded passages because the entire manuscript has effectively been re-written. 

Finally, we wish to express our sincere thanks for your dogged determination to ensure this paper was written and re-written to the best standard possible. While various issues conspired to frustrate our efforts, first author inexperience/challenge faced synthesising the original thesis, linked to adverse personal circumstances served to exacerbate the situation (though thankfully removed in time to work on this latest draft). We trust the outcome satisfies your considerable efforts on our behalf.

Reviewer 4 Report

None.

Author Response

We have addressed Reviewer 4's comments. We wish to thank him/her and all the reviewers for their insightful comments and for helping us to bring this manuscript to publication standard. Thank you again for all your support